# Effect of Paclobutrazol Application on Enhancing the Efficacy of Nitenpyram against the Brown Planthopper, *Nilaparvata lugens*

**DOI:** 10.3390/ijms241310490

**Published:** 2023-06-22

**Authors:** Xuhui Zhu, Qi Wei, Pinjun Wan, Weixia Wang, Fengxiang Lai, Jiachun He, Qiang Fu

**Affiliations:** State Key Laboratory of Rice Biology and Breeding, China National Rice Research Institute, Hangzhou 311401, China; 82101201157@caas.cn (X.Z.); wanpinjun@caas.cn (P.W.); wangweixia@caas.cn (W.W.); laifengxiang@caas.cn (F.L.); hejiachun@caas.cn (J.H.)

**Keywords:** *Nilaparvata lugens*, paclobutrazol, nitenpyram, synergetic effect, total phenolic, hydrogen peroxide

## Abstract

The brown planthopper (BPH), *Nilaparvata lugens*, is one of the most destructive rice pests in Asia. It has already developed a high level of resistance to many commonly used insecticides including nitenpyram (NIT), which is a main synthetic insecticide that is used to control BPH with a much shorter persistence compared to other neonicotinoid insecticides. Recently, we found that an exogenous supplement of paclobutrazol (PZ) could significantly enhance the efficacy of NIT against BPH, and the molecular mechanism underlying this synergistic effect was explored. The results showed that the addition of a range of 150–300 mg/L PZ increased the toxicity of NIT against BPH with the highest mortalities of 78.0–87.0% on the 16th day after treatments, and PZ could also significantly prolong the persistence of the NIT efficacies. Further investigation suggested that PZ directly increased the content of flavonoids and H_2_O_2_ in rice and increased the activity of polyphenol oxidase, which might be involved in the constitutive defense of rice in advance. Additionally, there was an interaction between PZ and BPH infestation, indicating that PZ might activate the host defense responses. Therefore, PZ increased the efficacy of NIT against the brown planthoppers by enhancing the constitutive and inducible defense responses of rice. Our study showed for the first time that PZ could contribute to improving the control effects of insecticides via inducing the defense responses in rice plants against BPH, which provided an important theoretical basis for developing novel pest management strategies in the field.

## 1. Introduction

Rice (*Oryza sativa*) is one of the most important staple food crops, feeding more than 50% of the global population [1]. The brown planthopper (BPH), *Nilaparvata lugens*, is a major destructive rice pest in East, Southeast and South Asia [2]. BPH ingests phloem sap through its stylet, absorbing rice nutrients and water and causing mechanical wounding; on the other hand, it spreads a variety of viruses, such as the rice ragged stunt virus [3,4]. BPH infestation causes leaf chlorosis, wilting and yield reduction in rice, causing economic losses to farmers [5]. For a long time, the use of chemical insecticides such as neonicotinoids has been a convenient and quick strategy for managing BPH in the field. However, the heavy application of insecticides leads to the continuous development of resistance in BPH, reducing the control efficiency and shortening the persistence of insecticides against BPH in rice paddy fields [6]. This led to the failure of insecticides such as imidacloprid to control the BPH; therefore, it is necessary to develop novel pest-management strategies.

Nitenpyram (NIT), a second-generation neonicotinoid insecticide, was developed and marketed by Takeda Agro in 1995 and introduced in China in 2007 [7,8]. It targets the acetylcholine receptor, and due to its good insecticidal activity, fast action, high efficiency and low toxicity to mammals, it is widely used against a variety of piercing–sucking mouthpart pests such as aphids, whiteflies and planthoppers [9]. Because of the high potential for BPH to develop insecticide resistance and the common use of NIT to control BPH, the resistance level of BPH increased slowly for a long time, but it is still only moderately resistant (10.0 ≤ resistance ratio < 100.0) at present [10,11]. This indicates that NIT can still be used as the main pesticide to control BPH. Currently, compared with other insecticides such as imidacloprid, which can suppress the BPH population for over 40 days in paddy fields, the main limiting factor for the application of NIT in the field is its short persistence, and the efficacy of NIT decreases rapidly 5 days after treatment [12,13]. Short persistence leads to multiple applications of NIT during the rice growth period, promoting the development of insecticide resistance in BPH and aggravating environmental pollution. Based on the previous screening results of our laboratory, we found that the control efficacy of the mixture of paclobutrazol (PZ) and NIT was significantly higher than that of NIT (Appendix A). However, there are few similar studies, and the mechanism of the synergistic effect is also not clear. We speculated that the synergistic effect of PZ on NIT may be achieved by altering the susceptibility of brown planthoppers to NIT.

PZ, a kind of plant gibberellin (GA) biosynthesis inhibitor, is one of the most widely used plant growth regulators in crop production and can prevent lodging, improve quality and increase the yield by 10.3–20.4% via seed treatment or foliar spray in rice production [14]. In addition to regulating rice growth and development, PZ has also been used to alleviate growth inhibition and reduce the oxidative damage caused by abiotic stress, such as flooding stress, salt stress and cadmium stress [15,16,17]. The physiological mechanisms of plants against biotic stress and abiotic stress are partly similar, and in recent years, there has been an increasing amount of literature on plant growth regulators enhancing rice defense against herbivores [18,19]. This indicates that PZ also has the potential to regulate plant defense.

The defense response of plants involves the participation of various enzymes and secondary metabolites [20]. Peroxidase (POD), polyphenol oxidase (PPO) and secondary metabolites including phenolic compounds and flavonoids are required in the regulation of rice growth and abiotic stress tolerance by PZ [21,22]. It is noteworthy that the above enzymes and secondary metabolites play an important role in regulating plant resistance to herbivorous insects [20]. In rice, phenolics and flavonoids are important secondary metabolites related to BPH resistance. Reducing the flavonoid level of rice can decrease the resistance of rice to brown planthoppers, resulting in an increase in the survival rate of brown planthoppers, while improving flavonoid levels leads to the opposite result [23,24]. The decrease in phenolic content also leads to the lower resistance of rice to brown planthoppers [25]. We speculated that the use of PZ had affected the defense of rice and thereby altered the susceptibility of the brown planthoppers to NIT. It is worth exploring whether these potential mechanisms are involved in the synergistic effect of PZ on NIT.

Nowadays, developing or searching for new synergistic agents has become a new strategy to reduce chemical pesticides and improve the effect [26]. Taking this line of thinking, we hoped to propose pesticide application pathways to reduce chemical pesticides and improve the effect and develop new strategies for insecticide resistance management in BPH by developing new synergistic agents. In addition, in order to satisfy the production needs of green agriculture, we limited the screening scope of synergists to plant growth regulators, which is beneficial for rice production. Through screening experiments, we found that PZ could be used as a synergistic agent for NIT. To accomplish our goal, we focused on a few main questions: (i) How long could the persistence of NIT be extended when mixed with PZ? (ii) Did PZ affect the susceptibility of brown planthoppers to NIT? (iii) What were the aspects of PZ regulating herbivore-induced defense responses in rice? In order to address the above questions, we first quantified the increment in the control efficiency and persistence of NIT after mixing with PZ. Then, we tested the effect of PZ on the susceptibility of brown planthoppers to NIT and studied the regulatory effect of PZ on the rice defense response to BPH. This study provides an important theoretical basis for the application of plant growth regulators to prolong the persistence of insecticides against pests.

## 2. Results

### 2.1. Synergistic Effect of Paclobutrazol on Nitenpyram against BPH

The lethal effects of PZ, NIT and PZ + NIT mixtures in the dipped method bioassay were calculated (Figure 1). On the 12th day after treatment, the mortality at 96 h of the BPH nymphs in the control group was 5.0%; the 200 and 300 mg/L PZ treatments and NIT treatment did not cause significant nymph mortalities at 96 h, which were only 8.0%, 10.0% and 5.0%, respectively. Mixing PZ and NIT increased the mortality rate. When the concentration of PZ in the PZ + NIT mixtures was higher than 100 mg/L, the mortalities at 96 h of the BPH nymphs significantly increased compared with the NIT group on the 12th day after treatment. When the concentration of PZ in the PZ + NIT mixtures was 150–300 mg/L, the PZ + NIT mixtures yielded the highest level of toxicity against the BPH with mortalities of 78.0–87.0% at 96 h on the 12th day after treatment.

### 2.2. Paclobutrazol Prolongs the Persistence of Nitenpyram against BPH

At different time points after treatment, bioassay experiments were conducted to investigate the synergistic characterization of PZ and NIT (Figure 2). At 0 d, the corrected mortalities at 24 h and 48 h of the PZ + NIT treatment were significantly higher than those of the NIT treatment by 30.0% and 20.0%, respectively; on the 6th day, the corrected mortalities at 24 h, 48 h and 96 h of the PZ + NIT treatment were significantly higher than those of the NIT treatment by 20.0%, 80.0% and 73.0%, respectively; on the 12th day, the corrected mortalities at 24 h, 48 h and 96 h of the PZ + NIT treatment were significantly higher than NIT treatment by 47.0%, 65.0% and 83.0%, respectively; on the 18th d, the corrected mortalities at 24 h, 48 h and 96 h of the PZ + NIT treatment were significantly higher than the NIT treatment by 38.0%, 43.0% and 53.0%, respectively; on the 24th d, there was no significant difference in mortalities between the PZ + NIT and NIT treatments at 24 h, 48 h and 96 h. These data indicated that PZ significantly prolonged the persistence of NIT.

### 2.3. Effect of Paclobutrazol on the Susceptibility of BPH to Nitenpyram

Toxicity bioassays at two time points were conducted to quantify the synergism of PZ on NIT. The results showed that PZ significantly increased the toxicity of NIT at 0 d and the 12th day after the dipping treatment, with synergism ratios of 15.1-and 4.94-fold, respectively (Table 1). The LC_50_ values of NIT and the PZ + NIT mixture on the third BPH nymphs were recorded as 7.55 mg/L and 0.50 mg/L, respectively, on the day after the dipping treatment, while the values were recorded as 525.37 mg/L and 106.30 mg/L, respectively, on the 12th day after the dipping treatment (Table 1).

### 2.4. Paclobutrazol Induces the Biosynthesis of Constitutive and Elicited Secondary Metabolites

After the BPH feeding for 96 h, the total phenolic content was 31.4% higher in the PZ-treated plants than in the control plants and 32.9% higher in the PZ + NIT-treated plants than that in the NIT-treated plants (Figure 3A). The three-way ANOVA results showed that the levels of the total phenolics in the rice stems were significantly enhanced by PZ, while the BPH and NIT treatments also had significant impacts (Table 2). It was surprising that the effect of the BPH infestation on the levels of the total phenolics varied between treatments, and the results showed that there was a significant interaction between the PZ treatment and BPH infestation (Table 2). Similarly, in the BPH infestation group and non-infested group, the levels of flavonoids were 16.8% and 20.6% higher in the PZ-treated plants than that in the control plants, respectively. The levels of flavonoids were 14.3% and 25.8% higher in the PZ + NIT-treated plants than that in the NIT-treated plants in the BPH infestation group and non-infested group, respectively. (Figure 3B). The three-way ANOVA results showed the flavonoid content was also significantly enhanced by PZ (Table 2). After the BPH feeding for 96 h, the level of malondialdehyde (MDA) in the rice stems was significantly enhanced by PZ, and the MDA content was 17.8% higher in the PZ-treated plants than that in the control plants and 16.6% higher in the PZ + NIT-treated plants than that in NIT-treated plants. In the non-infested group, PZ had no significant effect on the MDA content, and the results showed that the BPH infestation significantly enhanced the levels of MDA in rice (Figure 3C). The three-way ANOVA results showed that the MDA content was also significantly enhanced by PZ, and there was a significant interaction between the PZ treatment and BPH infestation (Table 2).

### 2.5. Paclobutrazol Alters the Biosynthesis of Constitutive and Elicited H_2_O_2_ and the Activities of Defense Enzymes

After the BPH feeding for 6 h, the contents of H_2_O_2_ were significantly enhanced by PZ in both the BPH infestation group and non-infested group, and the levels of H_2_O_2_ were 46.8% and 41.2% higher in the PZ-treated plants than those in the control plants in the BPH infestation group and non-infested group, respectively. The levels of H_2_O_2_ were 67.6% and 57.7% higher in the PZ + NIT-treated plants than those in the NIT-treated plants in the BPH infestation group and non-infested group, respectively (Figure 4A). At the two time points, the law of differences among different treatments was similar. After the BPH feeding for 12 h, in the non-infested group, the H_2_O_2_ content in the PZ-treated plants was 49.6% higher than that in the control plants, and there was no significant difference between the NIT-treated plants and PZ + NIT-treated plants. After the BPH feeding for 12 h, the H_2_O_2_ content in the PZ + NIT-treated plants were 52.7% higher than that in the NIT-treated plants, and there was no significant difference between the PZ and control treatments (Figure 4B). The three-way ANOVA results showed that the levels of H_2_O_2_ in the rice stems were significantly enhanced by PZ at both the 6 h and 12 h time points when the samples were collected, and the BPH infestation only significantly enhanced the levels of H_2_O_2_ in the rice stems at 6 h; furthermore, there was no significant interaction among the three factors. (Table 3).

The activities of POD and PPO were determined (Figure 5A,B). After the BPH feeding for 96 h, the POD activity of the PZ + NIT-treated plants were 15.9% higher than that of the NIT-treated plants, and there was no significant difference between the PZ treatment and the control in the BPH infestation group (Figure 5A). The three-way ANOVA results showed that the activity of the POD was not significantly enhanced by PZ, but there was a significant interaction between the PZ and NIT treatments (Table 3). PZ also influenced the activities of PPO, and the PPO levels of the PZ + NIT-treated plants were 83.8% and 74.0% higher than those of NIT-treated plants in both the BPH infestation group and non-infested group, respectively, while there was no significant difference between the PZ treatment and control (Figure 5B). The three-way ANOVA results showed that the activity of PPO was significantly enhanced by PZ, and there was a significant interaction between the PZ and NIT treatments (Table 3).

## 3. Discussion

Chemical control is an important part of integrated pest management (IPM). To reduce the environmental pollution caused by chemical insecticides and increase the economic benefits of farmers, improving the control efficiency of chemical insecticides and prolonging the persistence of insecticides have been current research hotspots [27]. In our study, we found that PZ could be used as a synergist of NIT against BPH, which had not been reported until now. In rice production, PZ is commonly used to decrease plant height, promote tillering occurrence and prevent lodging in the rice seedling stage, tillering stage and jointing stage, and its commonly used concentration range is 0 to 600 mg/L [14,28]. In our study, we demonstrated that the control efficiency of the PZ + NIT mixtures, whose concentration of PZ exceeded 100 mg/L, was significantly higher than that of NIT (Figure 1). More importantly, it was also demonstrated that the persistence of NIT can be prolonged to more than 18 days if mixed with PZ (Figure 2). Based on this phenomenon, we explored the potential mechanism of PZ as a synergist. Our results showed that PZ did not affect the content of NIT at different time points after dipping for 24 d, indicating that the synergistic effect of PZ might be achieved by affecting the rice physiology (Appendix A).

Ramoutar et al. [29] attempted to use several GA inhibitor plant growth regulators, including PZ, as synergists of bifenthrin to strengthen the toxicity to *Listronotus maculicollis*. The results showed that the synergistic ratio of PZ reached 2.2-fold, but the mortality of *L. maculicollis* was not significantly higher than that of the insecticide control. The determination method that the authors used was the topical application of 1 μL per insect dorsally to the intersegmental membrane between the prothorax and the elytra instead of feeding *L. maculicollis* with plants that were foliar sprayed with insecticides. This suggested that PZ might not directly enhance the efficacy of insecticides. However, we found that PZ enhanced the control effect of NIT after being applied to rice plants. This synergistic effect of PZ on NIT found in our study is unprecedented.

According to the above research results and previous studies, we speculated that there were two main reasons for the synergism caused by PZ: The first reason might be that PZ caused an increase in the content of pesticides in rice, and the second reason might be that the enhanced plant defense might lead to an increase in the susceptibility of brown planthoppers to pesticides. According to the dynamics of the NIT content of the two treatments, we eliminated the first possibility (Appendix A). Subsequently, when we found that the susceptibility of brown planthoppers to NIT was increased by PZ (Table 1), we focused on the effect of PZ on the rice defense response.

Plants respond to herbivores by synthesizing a variety of secondary metabolites that affect the feeding, growth and survival of herbivores [30]. According to Zhang et al. [25], silencing *OsSLR1* increased the content of phenolic acids, such as 4-OH-benzonic acid and 2,5-dihydroxybenzoic acid, in rice by 100.0%, which reduced the survival rate of BPH by 25.0% and the egg-hatching rate by 60.0%. In our experiment, we demonstrated that the application of PZ could significantly induce the accumulation of phenolic compounds (Table 2). The function of PZ in increasing plant phenolic content has been demonstrated. In rice, the treatment of seeds with PZ also significantly increased the total phenolic content by 30.0–80.0% [31]. Flavonoids are oligomeric phenolic compounds, and previous studies have shown that flavonoids are directly toxic to brown planthoppers; feeding artificial diets containing flavonoids can increase the mortality of brown planthoppers by 3.27-fold [32]. In our study, we concluded that the flavonoids content in the PZ-treated rice was significantly higher than that in the non-PZ-treated rice (Figure 3B, Table 2). Similar conclusions have been reported in other studies. It was indicated that PZ increased the content of flavonoids in soybean and potato by 14.4–19.7% [33,34]. These results both indicate that PZ enhanced the synthesis of herbivore-resistant substances in rice.

In the plant defense response system, H_2_O_2_ signaling is one of the critical pathways that regulate the rice response to herbivore damage [35,36,37,38]. Endogenous H_2_O_2_ levels increase after herbivorous insects, such as BPH and *Chilo suppressalis*, infest rice and thus elevate endogenous H_2_O_2_ levels in rice and cause a high resistance to herbivorous insects [38,39,40]. For example, spraying jasmonic acid enhances the resistance of *Arachis hypogaea* to *Helicoverpa armigera* by increasing its H_2_O_2_ level [41]. Our study demonstrated that the H_2_O_2_ level of rice plants is enhanced by PZ (Figure 4A,B, Table 3), and the difference in MDA content also supports this conclusion (Figure 3C). Additionally, POD and PPO can catalytically oxidize H_2_O_2_ and phenolic compounds into quinone, lignin and other insect-resistant secondary metabolites, so the two enzymes play an important role in plant resistance to herbivore insects [20,36]. Wang et al. [18] showed that enhanced POD activity induced phenolic polymerization and hindered *Sogatella furcifera* infestation. High PPO activity also prevented BPH from infesting rice [42]. Our study drew the conclusion that PZ enhanced the activity of rice defense enzymes (Figure 5C,D). The induction of POD by PZ was observed in *Catharanthus roseus* [43]. This indicated that PZ also enhanced the H_2_O_2_-mediated rice defense pathway. In addition, it could be inferred from the above results that the contents of quinone and lignin in the PZ-treated rice plants also increased, which might be the potential mechanism of the synergism of PZ.

Through the above inference, we believed that PZ directly increased the content of flavonoids and H_2_O_2_ in rice and increased the activity of PPO, which indicated that PZ might activate the constitutive defense of rice in advance (Table 2 and Table 3), increasing the resistance of rice to BPH. A BPH infestation further enhanced these changes, showing that these indices were independent of each other between the PZ and BPH infestation (Table 2 and Table 3). However, we also found that there was an interaction between the PZ and BPH infestation. As shown in Figure 3A, the significant impact of PZ on the total phenolic content only occurred in the infestation group, and the significant interaction between the PZ treatment and BPH infestation indicated that PZ might activate the induced defense response of rice (Table 2). Therefore, by enhancing the constitutive and induced defenses of rice, PZ increased the mortality of the BPH when the control efficacy of NIT decreased due to degradation and other factors, thus achieving the stable suppression of the BPH population over 18 days.

At present, the mechanisms by which PZ regulates H_2_O_2_ signaling and the total phenolics/flavonoids that increase the susceptibility of BPH to NIT are not clear. Xu et al. [44] hold the view that jasmonic acid (JA) signaling positively regulates rice resistance to brown planthoppers because the egg-hatching rate of BPH on the JA-deficient mutant was 50% higher than that of the wild-type plant. The JA signaling pathway affects the activities of POD and PPO and the synthesis of important secondary metabolites including phenolic compounds [44]. The GA and JA signaling pathways balance growth and defense by antagonism through the interaction of DELLA protein and JA ZIM-domain (JAZ) family proteins [45]. PZ can block the GA biosynthesis of plants by inhibiting the activity of *ent*-kaurene oxidase [46]. These findings suggest that PZ may have comparable physiological regulatory functions with JA, for example, with cadmium stress [17,47]. We speculate that the synergistic effect of PZ on NIT is related to the JA signaling pathway, but further research is needed to confirm this.

## 4. Materials and Methods

### 4.1. Plants and Insects

The rice variety used in this study was Zhongzheyou No.8, and the rice plants were used for the experiment at their 5th leaf stage. The rice plants were grown in a climate-controlled chamber (28 ± 1 °C, 14 h light phase, 70 ± 5% relative humidity). The cultivation of the rice plants followed the same procedure as described by Wang et al. [48].

A colony of BPH were originally collected from China National Rice Research Institute rice fields in Hangzhou, Zhejiang province, China, in 2020 and were reared on Taichung Native 1 under the conditions described above.

### 4.2. Chemicals

PZ (purity ≥ 95.0%) was obtained from Shanghai Macklin Biochemical Technology Co., Ltd., Shanghai, China. Commercial-grade nitenpyram (30% WDG) was obtained from Shanxi Huarong Kaiwei Biological Co., Ltd., Shanxi, China.

### 4.3. Procedure of Chemical Solution and Plant Treatments

The whole rice plant dipping assay method was adopted to assess the chemical toxicity of the chemical mixtures. First, PZ was dissolved in acetone at a concentration of 10,000 mg/L, and NIT was dissolved in distilled water at a concentration of 2500 mg/L. Both stock solutions were mixed to prepare the final concentrations of 80 mg/L NIT, which could cause approximately 50% of the brown planthoppers to die on the 7th day postexposure to 50 to 300 mg/L PZ (lower than the highest recommended doses). Meanwhile, a 3% (*v*/*v*) acetone solution, 200 and 300 mg/L paclobutrazol solution and 80 mg/L NIT solution were prepared as the controls. The chemical treatment procedures of the experimental plants were as follows: The redundant tillers and senescing leaves were pruned (only the main stem was reserved) after the rice plants were cleaned and dried. Then, these plants were inverted and dipped carefully for 30 s into a 2.5 L graduated cylinder containing the mixed solution and were air dried at room temperature for 1 h. Finally, they were transferred to the climate-controlled chamber for further experiments.

### 4.4. Bioassay

All the rice plants exposed to the different concentrations of chemical mixtures were individually confined within a double-pass plastic cylinder (2.7 cm in diameter, approximately 8 cm high) into which 20 third instar BPH nymphs were transferred. The opening was sealed around the rice stem with a piece of sponge to prevent the escape of the insects tested. The nymph mortality rate per plant of all treatments was recorded at 24, 48 and 96 h postinfestation (details about the time point of the bioassay can be seen in Figure 1 and Figure 2). The rice plants that were not exposed to chemical reagents were the blank controls. According to the results of the concentration screening experiment, 200 mg/L was selected as the subsequent experimental concentration of PZ.

### 4.5. Determination of Total Secondary Metabolites, H_2_O_2_ and Defense Enzyme Activity

The sample collection method referred to Wang et al. [18] with some modifications. In brief, plants were randomly assigned to the blank control (adding an equal volume of acetone), 200 mg/L PZ (synergist control), NIT (insecticide control) and PZ + NIT groups. Half of the plants in each treatment were infested with 20 third instar BPH nymphs, and the others were used as a noninfestation control. The same method described in ‘Bioassay’ was used to make the nymphs of the brown planthopper feed on rice. Approximately 8 cm basal stems from individual plants were harvested at the indicated time points after the start of the BPH nymph infestation. These specimens were ground in liquid nitrogen and stored in a −80 °C refrigerator for subsequent analyses. Herein, the samples collected at 6 h and 12 h were used to determine the content of H_2_O_2_, and the rest of the indicators were determined by using samples collected at 96 h.

#### 4.5.1. Secondary Metabolite Analysis

Secondary metabolites are important substances for plants to resist herbivorous insects, such as phenolics and flavonoids, so the content of these two substances were determined. Total phenolics were extracted and quantified by using the method described in [49]. In brief, 100 mg of the specimen powder was homogenized with 2 mL of ice-cold 95% (*v*/*v*) methanol. The specimen mixture was incubated in a 2 mL microtube at room temperature for 48 h in the dark. Then, the specimen mixture was centrifuged (13,000× *g* for 5 min at room temperature) and the supernatant was collected. Then, 100 μL of each specimen supernatant and 200 μL of the 10% Folin–Ciocalteu reagent were mixed and vortexed thoroughly. Finally, 800 μL of 700 mM Na_2_CO_3_ was added into the tube. The assay tubes were incubated at room temperature for 2 h and the absorbance at 765 nm was read. The standard curve of gallic acid was established to determine the total phenolic concentration of the sample solution C_P_ (gallic acid is a typical phenolic compound, generally considered as being 100% total phenolic). Absorbance was measured by using an NP80 UV–Vis spectrophotometer (Tecan Infinite M200 PRO). The calculation of the total phenolic content was based on the following formula:$$\mathrm{Total}\;\mathrm{phenolic}\;\mathrm{content}=\frac{C_{p}\times 1}{fresh\;weight\;\left(FW\right)}$$

Flavonoids were extracted and quantified by using the method described in [50]. Briefly, 200 mg of the specimen powder was homogenized with 4 mL of 70% (*v*/*v*) ethanol and was extracted in an ultrasonic cleaner (500 W, 40 kHZ, 60 °C) for 15 min. The specimen mixture was centrifuged (3000× *g* rpm for 20 min at room temperature) and the supernatant was collected. Then, 2 mL of each specimen supernatant, 400 μL of 5% (*w*/*v*) NaNO_2_, 400 μL of 10% (*w*/*v*) Al(NO_3_)_3_, 4 mL of 4% (*w*/*v*) NaOH and 3.2 mL of 70% (*v*/*v*) ethanol were mixed. Finally, the absorbance at 505 nm was read. The standard curve of rutin was established to determine the flavonoids concentration of the sample solution C_F_ (rutin is a flavonoid, generally considered as being 100% a flavonoid). The calculation of the flavonoids content was based on the following formula:$$\mathrm{Flavonoids}\;\mathrm{content}=\frac{C_{F}\times 20}{FW}$$

MDA was extracted and quantified by using the thiobarbituric acid reaction method described by Gu et al. [51]. In brief, 1000 mg of sample powder was homogenized in 10 mL of a HEPES buffer (50 mM; pH 7.8; and containing 1 mM EDTA, 1 mM ascorbic acid, 1 mM GSH, 5 mM MgCl_2_, 1 mM DTT and 10% glycerol) and centrifuged (12,000× *g* for 30 min, 4 °C). The supernatant was collected as the crude enzyme solution. Four milliliters of a trichloroacetic acid (202.5 g/L)–thiobarbituric acid (5 g/L) mixed solution and 2 mL of an enzyme solution were mixed in a 15 mL centrifugal tube, heated (100 °C) for 20 min and then centrifuged (4000× *g* for 10 min at room temperature). The absorbance at 450 nm, 532 nm and 600 nm was read. The MDA content was calculated by using the following formula:$$\mathrm{MDA}\;\mathrm{content}=\frac{\left[6.452\times (D_{532}-D_{600}\right)-0.559\times D_{450}]\times 5}{FW}$$

#### 4.5.2. H_2_O_2_ Analysis

It is known that the H_2_O_2_ pathway positively regulates the resistance of rice to BPH, so the levels of H_2_O_2_ were determined in this study. The H_2_O_2_ concentration was determined using a hydrogen peroxide assay kit (Beijing Solarbio Science & Technology Co., Ltd., Cat# BC3595, Beijing, China) according to the manufacturer’s instructions. Briefly, 100 mg of the sample powder was homogenized in 1 mL of acetone and centrifuged (8000× *g* for 10 min, 4 °C). The supernatant was used for determination. After the reaction, the absorbance at 415 nm of all the solutions was read. The H_2_O_2_ content was determined according to the absorbance of the H_2_O_2_ standard curve (D_s_), and the H_2_O_2_ content was calculated by using the following formula:$$H_{2}O_{2}\;\mathrm{content}=\frac{2\times (D_{\mathrm{test}}-D_{s})}{FW}$$

#### 4.5.3. Defense Enzyme Activity

The POD levels were determined by using the method described in [51] with slight modifications. Briefly, 100 μL of a crude enzyme solution (same as that in the MDA analysis) was rapidly reacted with a mixture of 2 mL of acetate buffer (100 mM, pH 5.4), 800 μL of 0.25% (*v*/*v*) *o*-methoxy-phenol and 100 μL of 0.075% (*w*/*v*) H_2_O_2_. The change in absorbance at 470 nm was recorded over 3 min. The POD activities were expressed as U·mg^−1^ protein, and one unit of POD activity was defined as the amount of enzyme that caused a change of 0.1 in absorbance per minute. The protein concentration of the crude enzyme solution was determined as described by Bradford [52].

The PPO levels were determined by using the method described in [53] with slight modifications. In summary, 100 mg of sample powder was homogenized in 1 mL of a PBS buffer (50 mM, pH 6.0, containing 1% PVP) and centrifuged (12,000× *g* for 30 min, 4 °C). The supernatant was collected as the crude enzyme solution, and 10 μL of the crude enzyme solution was rapidly reacted with a mixture (preheated at 30 °C for 5 min) of 170 μL of the PBS buffer (50 mM, pH 6.0) and 20 μL of 100 mM catechol. The change in absorbance at 410 nm was recorded over 3 min. The PPO activities were expressed as U·mg^−1^ FW, and one unit of PPO activity was defined as the weight of the sample that caused a change of 0.1 in the absorbance per minute.

### 4.6. Data Analysis

The LC_50_ and its 95% confidence limits, slope, chi-square value (χ^2^) and degree of freedom (*df*) for each treatment were determined via a probit analysis that was conducted with SPSS 26 software (IBM SPSS Statistics, Redmond, WC, USA) as described by Cai et al. [54]. The toxicity comparison was determined by whether the confidence limits overlapped. The synergism ratio was calculated by dividing the LC_50_ without PZ by the LC_50_ with PZ, and the data on the levels of the total phenolic compounds, flavonoids, MDA, H_2_O_2_, POD and PPO for the different treatments were analyzed by using one-way and three-way ANOVAs followed by Tukey’s multiple range tests (significant difference: *p* < 0.05). Differences between the two treatments were analyzed by using Student’s *t* tests. The statistical analysis was performed by using the SPSS 26 software, and the graphs were drawn by using GraphPad Prism (version 8.4; GraphPad Software, San Diego, CA, USA).

## 5. Conclusions

In summary, our study proved for the first time that PZ could be used as a synergist of NIT. We demonstrated that adding PZ increased the susceptibility of *N. lugens* to NIT, and this phenomenon might be related to PZ increasing the levels of H_2_O_2_, POD, PPO, total phenolics and flavonoids in rice. These research findings promote the reduction in chemical pesticides, improvement in the effect of chemical pesticides and development of new strategies for the insecticide-resistance management of BPH. However, there is still a lot of work to be conducted in the future. Firstly, the molecular mechanism that PZ regulating rice resistance against brown planthoppers is still not clear, and analyzing this mechanism contributes to the development of synergists and the breeding of insect-resistant varieties. Additionally, the physiological change mechanism of brown planthoppers caused by PZ is also one of the future research directions. Thirdly, relevant field experiments also need to be conducted, and the screening of other plant growth regulators that can serve as insecticide synergists needs to be further studied.

## Figures and Tables

**Figure 1 ijms-24-10490-f001:**
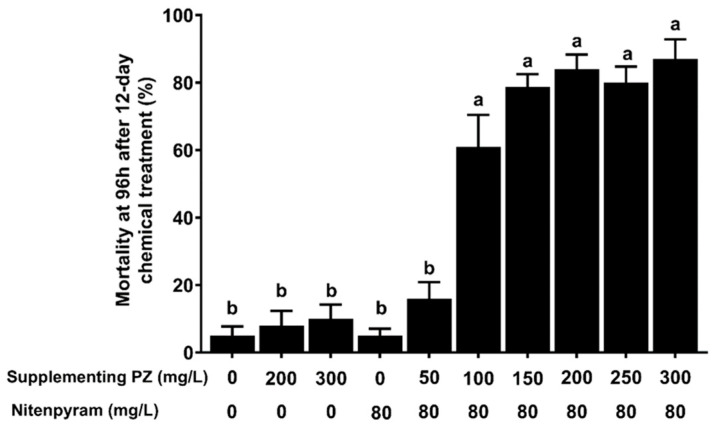
Toxicity of the chemical mixtures containing different PZ concentrations against BPH nymphs. The bioassay was conducted on 12th day after the host plants were dipped and the mortalities were recorded after 96 h. Data are presented as the means ± S.E. for five independent replicates. The bars with different small letters indicate significant differences among different treatments (*p* < 0.05, Tukey’s multiple range test).

**Figure 2 ijms-24-10490-f002:**
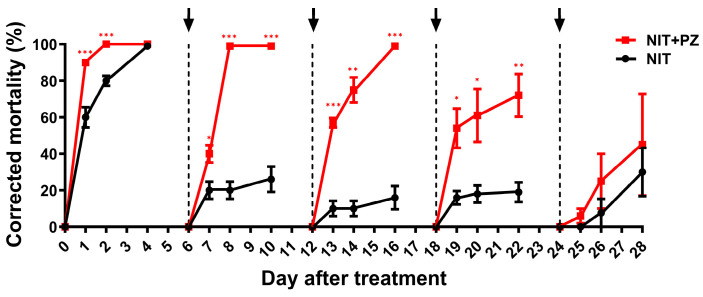
Toxicity comparison of NIT and its mixture containing PZ against BPH nymphs within 4 weeks after chemical treatments. The bioassays were conducted at 0, 6, 12, 18 and 24 d after rice plants were dipped, and the mortalities were recorded at 24, 48 and 96 h after the beginning of bioassays. The dotted lines and arrows represent the beginning of the independent test bioassays. Data are presented as the mean ± S.E. for five independent replicates. Asterisks indicate significant differences between the treatments of NIT and NIT + PZ at the same time point (* *p* < 0.05; ** *p* < 0.01; *** *p* < 0.001; Student’s *t*-test).

**Figure 3 ijms-24-10490-f003:**
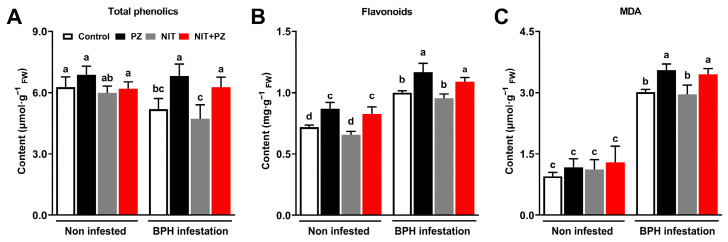
The level of total phenolic, flavonoids and malonaldehyde (MDA) in different treatment plants. Mean levels (+SE, *n* = 5) of total phenolic (**A**), flavonoids (**B**) and MDA (**C**) in stem of rice plants at 96 h after infestation under different treatments for 12th day. The bars with different small letters indicate significant differences among different treatments (*p* < 0.05, Tukey’s multiple range test).

**Figure 4 ijms-24-10490-f004:**
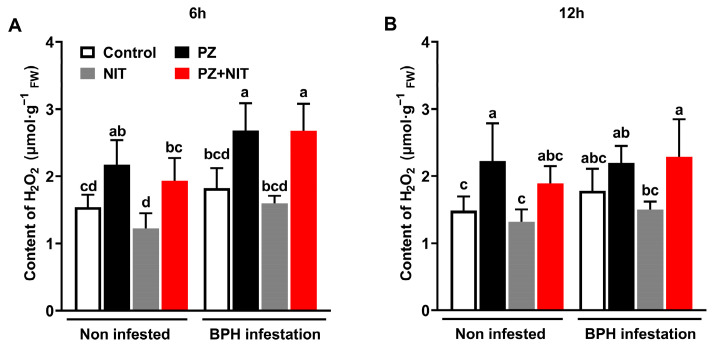
The level of H_2_O_2_ in different treatment rice plants. Mean levels (+SE, *n* = 5) of H_2_O_2_ in stem of rice plants at 6 (**A**) and 12 (**B**) h after infestation under different treatments at 12th day. The bars with different small letters indicate significant differences between different treatments (*p* < 0.05, Tukey’s multiple range test).

**Figure 5 ijms-24-10490-f005:**
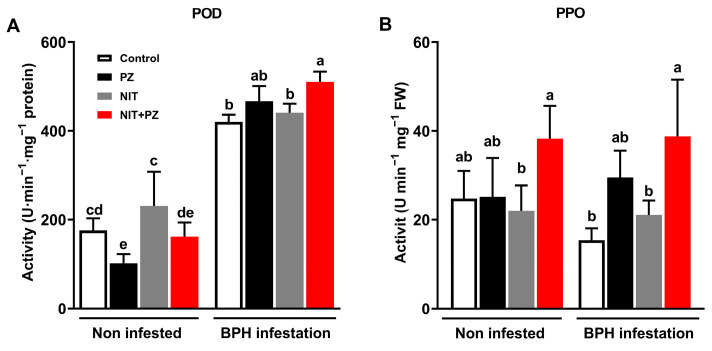
Mean levels (+SE, *n* = 5–6) of POD (**A**) and PPO (**B**) in stem of rice plants at 96 h after infestation under different treatments at 12th day. The bars with different small letters indicate significant differences between different treatments (*p* < 0.05, Tukey’s multiple range test).

**Table 1 ijms-24-10490-t001:** The toxicity of NIT and NIT combined with PZ to BPH after 96 h.

Day after Treatment	Treatment	Slope ± SE	LC_50_ (mg/L) 95% C.L. ^a^	χ^2^ (*df*)	SR ^b^
0	NIT	2.01 ± 0.29	7.55 (6.50–8.69)	5.46 (4)	15.1
	PZ + NIT	2.39 ± 0.41	0.50 (0.42–0.58)	0.66 (7)	
12	NIT	1.21 ± 0.19	525.37 (326.95–1076.06)	15.51 (5)	4.94
	PZ + NIT	1.46 ± 0.20	106.30 (68.89–157.17)	16.08 (5)	

^a^ Confidence limits. ^b^ Synergistic ratio = LC_50_ of NIT + PZ/LC_50_ of NIT.

**Table 2 ijms-24-10490-t002:** Three-way ANOVA results of phenolic, flavonoid and MDA of different treatments.

Treatment	Phenolic Content/μmol·g^−1^ FW	Flavonoids Content/mg·g^−1^ FW	MDA Content/mg·g^−1^ FW
BPH	0.000 ***	0.000 ***	0.000 ***
NIT	0.002 **	0.000 ***	0.62
PZ	0.000 ***	0.000 ***	0.000 ***
BPH×NIT	0.899	0.611	0.102
BPH×PZ	0.000 ***	0.76	0.022 *
NIT×PZ	0.419	0.812	0.748
BPH×NIT×PZ	0.592	0.296	0.984

* *p* < 0.05; ** *p* < 0.01; *** *p* < 0.001.

**Table 3 ijms-24-10490-t003:** Three-way ANOVA results of H_2_O_2_, POD activity and PPO activity of different treatments.

BPH Treatment	H_2_O_2_ 6 h after BPH Infestion/μmol g^−1^ FW	H_2_O_2_ 12 h after BPH Infestion/μmol g^−1^ FW	POD Activity/U	PPO Activity/U
BPH	0.000 ***	0.063	0.000 ***	0.556
NIT	0.054	0.126	0.001 **	0.010 *
PZ	0.000 ***	0.000 ***	0.102	0.000 ***
BPH × NIT	0.416	0.485	0.443	0.62
BPH × PZ	0.138	0.805	0.198	0.109
NIT × PZ	0.449	0.638	0.000 ***	0.042 *
BPH × NIT × PZ	0.701	0.221	0.940	0.188

* *p* < 0.05; ** *p* < 0.01; *** *p* < 0.001.

## Data Availability

Not applicable.

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
