# Peer review of "Effect of Paclobutrazol Application on Enhancing the Efficacy of Nitenpyram against the Brown Planthopper, Nilaparvata lugens"

_ijms, 2023, doi:10.3390/ijms241310490_

Round 1

Reviewer 1 Report

The manuscript entitled "Effects of paclobutrazol application on enhancing the efficacy of nitenpyram against the brown planthopper, Nilaparvata lugens" aims to explore the synergism between paclobutrazol and nitenpyram and their efficacy in the resistance against planthopper. Although the paper seems interesting, some questions/comments are stated below and should be addressed before considering the publication of this manuscript in the International Journal of Molecular Sciences:

1. Please consider including a list of abbreviations commonly used along the manuscript.

2. Lines 60-62: In these sentences of the introduction, the authors stated conclusions of the unpublished data. Why are these data not published at least in supplementary material? Is it possible to include them to support these statements?

3. Lines 76-78: Is there a specific reason for including this conclusion of the present study in the introduction? Please consider including it in the results or discussion sections.

4. Lines 94-99: Please clarify the objectives of the study in these paragraph of the Introduction. 

5. All figures and tables should be placed immediately after they are first mentioned in the manuscript. Please revise it carefully.

6. Figure 2: The graph is confusing considering the days of the treatment and respective hours cited in the legend. Please revise it.

7. Table 1: The statistical analysis is missing in the table. Please include the statistics.

8. Tables 2 and 3: Please indicate in the footnotes the meaning of *, **, and ***.

9. Methods: Please include the following information for each assay: i) how were the results expressed for each assay?; and ii) what were the controls and/or blanks used? Please indicate it.

10. The conclusion may benefit from discussing some future perspectives on the research topic studied. 

11. References need to be carefully revised following the journal guidelines.

Minor comments:

- Line 18: Define the abbreviation "PZ".

- Line 21: Define the abbreviation "PPO".

- Line 62: Correct "post-treatment".

- Lines 73-76: Please include references to support these statements.

- Lines 82-83: Add at least one reference to support this statement.

- Lines 101, 114, 128, 137, and 159: Please consider using the full names instead of abbreviations in the respective subtitles. 

- Lines 116-124: Please replace "6 d" by "6th day". Consider revising it for all the similar mentions throughout the manuscript.

- Line 132: Please define "LC50" and format "50" in subscript (i.e., below the line).

- Lines 160, 364, 396, 405, and 423: Delete the points at the end of the subtitles please.

- Line 190: Delete "Fig.1" after "Figure 1" in the legend. 

- Line 192: Define the abbreviation "S.E.".

- Figure 3: Define the abbreviation "FW".

- Lines 222 and 287: Please reformat properly "H2O2".

- Line 307: Please define the abbreviation "CK".

- Line 310: Please define the abbreviations "DELLA" and "ZIM".

- Lines 318-423: Add spaces between different subsections regarding methods.

- Line 328: Define the abbreviation "WDG".

- Line 335: Correct "post-exposure".

- Line 350: Correct "post-plant".

- Line 375: What were the concentrations of gallic acid tested for TPC? Please include this information.

- Line 384: What were the concentrations of rutin tested for TFC? Please include this information.

- Line 388: Please define the abbreviations "HEPES" and "EDTA".

- Line 389: Please define the abbreviations "GSH" and "DTT".

- Lines 394-395: Please revise the way that the formula is presented and reformat it according to the journal guidelines.

- Line 409: Format "o" from "o-methoxy-phenol" in italic letter.

- Line 415: Define the abbreviation of "PBS".

- Line 416: Define the abbreviation of "PVP".

- Line 430: Format "p" from "p < 0.05" in italic letter.

Author Response

Response to Reviewer 1 Comments

Point 1: Please consider including a list of abbreviations commonly used along the manuscript.

Response 1: We have revised it and added the abbreviation list at the end of this article (Page 12, Line 497).

Point 2: Lines 60-62: In these sentences of the Introduction, the authors stated conclusions of the unpublished data. Why are these data not published at least in supplementary material? Is it possible to include them to support these statements?

Response 2: Actually, due to its significant control effect, paclobutrazol is picked out from more than 15 plant growth regulators or other compounds through a series of insecticide bioassays. We have attached the related unpublished data as supplementary file (Figure S1).

Point 3: Lines 76-78: Is there a specific reason for including this conclusion of the present study in the introduction? Please consider including it in the results or discussion sections.

Response 3: Thanks for your comments. We have removed the related conclusive sentences from Introduction part and described them as being part of the Discussion (Page 7, Lines 240-243).

Point 4: Lines 94-99: Please clarify the objectives of the study in these paragraphs of the Introduction.

Response 4: Thanks for the suggestions. I have made some modifications to what you have pointed out at the last paragraph of the introduction, the added content is as follows: Nowadays, developing or searching for new synergistic agents has become a new strategy to reduce chemical pesticides and improve the effect [26]. Taking this line of thinking, we hoped to propose pesticide application pathways for reducing chemical pesticides and improving the effect and developing new strategies for insecticides re-sistance management in BPH by developing new synergistic agents. In addition, in or-der to satisfy the production needs of green agriculture, we limited the screening scope of synergists to plant growth regulators, which was beneficial to rice production. Through screening experiments, we found that PZ could be used as a synergistic agent for nitenpyram. To accomplish our goal, we focused on a few main questions: (i) how long could the persistence of nitenpyram be extended when mixed with PZ? (ii) did PZ affect the susceptibility of brown planthopper to nitenpyram? (iii) what were the as-pects of PZ regulating herbivore-induced defense responses in rice. In order to answer the above questions, we first quantified the increment in the control efficiency and persistence of nitenpyram after mixing with PZ. (Pages 2-3, Lines 90-103).

Point 5: All figures and tables should be placed immediately after they are first mentioned in the manuscript. Please revise it carefully.

Response 5: We have revised them according to your suggestions and the guidelines of Journal (Page 3, Line 122; Page 4, Line 141; Page 4, Lines 157-159; Page 5, Line 181; Page 5, Lines 187-188; Page 6, Lines 219-229).

Point 6: Figure 2: The graph is confusing considering the days of the treatment and respective hours cited in the legend. Please revise it.

Response 6: We have made modifications to Figure 2 (Page 4, Line 141).

Point 7: Table 1: The statistical analysis is missing in the table. Please include the statistics.

Response 7: Thanks for your attention. Gernerally, the toxicity comparison is determined by whether the confidence limits overlap. We have added relevant descriptions in the Materials and Methods section, and also listed some previous research as evidence below:

Wei, Q.; Mu, X.-C.; Yu, H.-Y.; Niu, C.-D.; Wang, L.-X.; Zheng, C.; Chen, Z.; Gao, C.-F., Susceptibility of Empoasca vitis (Hemiptera: Cicadellidae) populations from the main tea-growing regions of China to thirteen insecticides. Crop Protection 2017, 96, 204-210.

Wheeler, M. W.; Park, R. M.; Bailer, A. J., Comparing median lethal concentration values using confidence interval overlap or ratio tests. Environ Toxicol Chem 2006, 25, (5), 1441-4.

Point 8: Table 1: The statistical analysis is missing in the table. Please include the statistics.

Response 8: We have checked and revised them. (Page 5, Line 188; Page 6, Line 229).

Point 9: Methods: Please include the following information for each assay: i) how were the results expressed for each assay?; and ii) what were the controls and/or blanks used? Please indicate it.

Response 9: Thanks for your comments. For the first question, we have supplemented the information about data processing or parameters calculation (Page 10, Line 395; Pages 10, Line 407; Page 10, Line 418; Page 10, Line 429); For the second question, in fact, we used three different controls in our study, namely blank control, synergist control (PZ treatment), and insecticide control (NIT treatment), the corresponding changes have been made in the 4.5 subsection of Materials and Methods section (Page 9, Lines 369-372).

Point 10: The conclusion may benefit from discussing some future perspectives on the research topic studied.

Response 10: Thanks for the suggestions. As you mentioned, we have added these sentences at the end of the Conclusion section: These research findings promote reducing chemical pesticides, improving the effect and developing new strategies for insecticides resistance management in BPH. But there is still a lot of work to be done in the future. Firstly, the molecular mechanism of PZ regulation of rice resistance against brown planthopper is still not clear, analyzing this mechanism contributes to the development of synergists and the breeding of insect resistant varieties. Besides, the physiological changes mechanism of brown planthopper caused by PZ is also one of the future research directions. Thirdly, relevant field experiments also need to be conducted and the screening of other plant growth regulators that can serve as insecticide synergists need to be further studied (Page 11, Lines 466-475).

Point 11: References need to be carefully revised following the journal guidelines.

Response 11: We have checked and revised them following the journal guidelines.

Point 12: Line 18: Define the abbreviation "PZ".

Response 12: We have revised it and added the abbreviation at the end of this article (Page 12, Line 497).

Point 13: Line 21: Define the abbreviation "PPO".

Response 13: We have revised it and added the abbreviation at the end of this article (Page 12, Line 497).

Point 14: Line 62: Correct "post-treatment".

Response 14: Thanks for your attention. In our modification process, this word has been deleted.

Point 15: Lines 73-76: Please include references to support these statements.

Response 15: Thanks for the suggestions. We have removed this conclusive statement and replace the relevant description as following: We speculated that the use of PZ had affected the defense of rice, thereby altering the susceptibility of brown planthoppers to nitenpyram. It is worth exploring whether these potential mechanisms are involved in the synergistic effect of PZ on nitenpyram. (Page 2, Lines 86-89).

Point 16: Lines 82-83: Add at least one reference to support this statement.

Response 16: A reference has been added. The corresponding reference numbers have also been changed (Page 2, Line 77; Page 2, Line 79; Page 2 Line 81; Page 12, Lines 546-553)

Point 17: Lines 101, 114, 128, 137, and 159: Please consider using the full names instead of abbreviations in the respective subtitles.

Response 17: A reference has been added. Changed accordingly. (Page 2, Line 110; Page 2, Line 128; Page 2 Line 149; Page 2, Line 160; Page 2, Line 189)

Point 18: Lines 116-124: Please replace "6 d" by "6th day". Consider revising it for all the similar mentions throughout the manuscript.

Response 18: We have checked and revised them. (Page 3, Line 124; Page 3, Line 132; Page 3, Line 134; Page 4, Line 152; Page 4, Line 156;)

Point 19: Line 132: Please define "LC50" and format "50" in subscript (i.e., below the line).

Response 19: Changed accordingly (Page 4, Line 153).

Point 20: Lines 160, 364, 396, 405, and 423: Delete the points at the end of the subtitles please.

Response 20: Thanks for the suggestions. I have tried to modify them but them couldn't change be changed under the special template provided by the Journal.

Point 21: Line 190: Delete "Fig.1" after "Figure 1" in the legend.

Response 21: Changed accordingly. (Page 3, Line 123)

Point 22: Line 192: Define the abbreviation "S.E.".

Response 22: We have revised it and added the abbreviation list at the end of this article (Page 12, Line 497).

Point 23: Figure 3: Define the abbreviation "FW".

Response 23: We have revised it and added the abbreviation list at the end of this article (Page 12, Line 497).

Point 24: Lines 222 and 287: Please reformat properly "H2O2".

Response 24: Changed accordingly (Page 6, Line 228; Page 8, Line 296).

Point 25: Line 307: Please define the abbreviation "CK".

Response 25: Thanks for the suggestions. I'm sorry that when I cited this reference, I didn't specify what CK represents. The "CK" you pointed out in the article has been changed to "wild-type plant" (Page 8, Line 316).

Point 26: Please define the abbreviations "DELLA" and "ZIM".

Response 26: Thanks for the suggestions. This abbreviation of "ZIM" has been included in the supplementary list (Page 12, Line 497); As for DELLA, in fact, DELLA is not an abbreviation, but a gibberellin signaling pathway-related protein that contains specific domains composed of aspartic acid (D), glutamic acid (E), leucine (L), leucine (L), and alanine (A), I have added DELLA's explanation to the list of abbreviations (Page 12, Line 497).

Point 27: Lines 318-423: Add spaces between different subsections regarding methods.

Response 27: Changed accordingly.

Point 28: Line 328: Define the abbreviation "WDG".

Response 28: We have revised it and added the abbreviation list at the end of this article (Page 12, Line 497).

Point 29: Line 335: Correct "post-exposure".

Response 29: Changed accordingly (Page 9, Line 346).

Point 30: Line 350: Correct "post-plant".

Response 30: Changed accordingly (Page 9, Line 362).

Point 31: Line 375: What were the concentrations of gallic acid tested for TPC? Please include this information.

Response 31: Thanks for your comments. Gallic acid is a typical phenolic compound, generally considered as 100% TPC. The corresponding information has been added to Materials and Methods section (Page 10, Lines 391-392).

Point 32: Line 384: What were the concentrations of rutin tested for TFC? Please include this information.

Response 32: Thanks for your comments. Rutin is a flavonoid, generally considered as 100% TFC. The corresponding information has been added to Materials and Methods section (Page 10, Lines 404-405).

Point 33: Line 388: Please define the abbreviations "HEPES" and "EDTA".

Response 33: We have revised it and added the abbreviation list at the end of this article (Page 12, Line 497).

Point 34: Line 389: Please define the abbreviations "GSH" and "DTT".

Response 34: We have revised it and added the abbreviation list at the end of this article (Page 12, Line 497).

Point 35: Lines 394-395: Please revise the way that the formula is presented and reformat it according to the journal guidelines.

Response 35: The formula has been modified accordingly (Page 10, Line 418).

Point 36: Line 409: Format "o" from "o-methoxy-phenol" in italic letter.

Response 36: Changed accordingly (Page 10, Line 435).

Point 37: Line 415: Define the abbreviation of "PBS".

Response 37: We have revised it and added the abbreviation list at the end of this article (Page 12, Line 497).

Point 38: Line 416: Define the abbreviation of "PVP".

Response 38: We have revised it and added the abbreviation list at the end of this article (Page 12, Line 497).

Point 39: Line 430: Format "p" from "p < 0.05" in italic letter.

Response 39: Changed accordingly (Page 11, Line 458).

Reviewer 2 Report

This is a very interesting work that demonstrates that the addition of paclobutrazol (PZ) enhances the efficacy of nitenpyram (NIT) against the brown planthopper (BPH). According to the results, PZ increases susceptibility of BPH to NIT by increasing the levels of several compounds (H2O2, POD, PPO, phenolics and flavonoids) in rice plants.

The article is well structured and presented. The research outcomes are relevant.

The Results are well presented. The statistical analysis is consistent.

The Discussion is good.

Materials and Methods are correct.

The Conclusion is consistent with the results.

1. Lines 149-150 (page 3) -> "The three-way ANOVA results showed the flavonoids content of was also significantly enhanced by PZ (Table 2)." Maybe, you want to say: "...showed the flavonoids content was also..."

2. Line 151 (page 4) -> you mention MDA for first time. Indicate for this first time that MDA is malonaldehyde, please. When using an abbreviation for a chemical compound for the first time, please provide the full name of the compound.

3. Line 190 (page 5) -> "Figure1. Fig.1 Toxicity of the..." 

4. Why do you put the Figures and Tables in a separate heading? Is not better to place them as the text needs them? 

5. line 287 (page 8) -> "H2O2" use subscripts.

6. line 350 (page 9) -> "...postplant infestation (details about the time point of bioassay can be seen in Figs)..." Please specify which Figures.

7. line (426) (page 10) -> "...(IBMSPSS Statistics, Redmond, WC, USA) as described by Cai, et al. [32]. The..." Leave a space between the two words.

8. Please, review citation number 1.

9. Could you add a sentence or a paragraph about future research lines in the Conclusions Section?

Author Response

Response to Reviewer 2 Comments

Point 1: Lines 149-150 (page 3) -> "The three-way ANOVA results showed the flavonoids content of was also significantly enhanced by PZ (Table 2)." Maybe, you want to say: "...showed the flavonoids content was also...".

Response 1: Very sorry about that, this sentence has been revised to “The three-way ANOVA results showed the flavonoids content was also significantly enhanced by PZ (Table 2)” (Page 4, Lines 172-173).

Point 2: Line 151 (page 4) -> you mention MDA for first time. Indicate for this first time that MDA is malonaldehyde, please. When using an abbreviation for a chemical compound for the first time, please provide the full name of the compound.

Response 2: We have provided the full name and added the abbreviation at the end of this article (Page 4, Line 174; Page 12, Line 497).

Point 3: Line 190 (page 5) -> "Figure1. Fig.1 Toxicity of the...".

Response 3: The original legend has been changed to: “Figure 1. Toxicity of the chemical mixtures containing different PZ concentrations against BPH nymphs.” (Page 3, Line 123).

Point 4: Why do you put the Figures and Tables in a separate heading? Is not better to place them as the text needs them?

Response 4: We have revised them according to your suggestions and the guidelines of journal (Page 3, Line 122; Page 4, Line 141; Page 4, Lines 157-159; Page 5, Line 181; Page 5, Lines 187-188; Page 6, Lines 219-229).

Point 5: line 287 (page 8) -> "H2O2" use subscripts.

Response 5: Changed accordingly (Page 6, Line 228; Page 8, Line 296).

Point 6: line 350 (page 9) -> "...postplant infestation (details about the time point of bioassay can be seen in Figs)..." Please specify which Figures.

Response 6: This sentence has been changed to: “The nymph mortality rate per plant of all treatments were recorded at 24, 48, 96 h post-plant infestation (details about the time point of bioassay can be seen in Fig 1&2)” (Page 9, Lines 361-362).

Point 7: line (426) (page 10) -> "...(IBMSPSS Statistics, Redmond, WC, USA) as described by Cai, et al. [32]. The..." Leave a space between the two words.

Response 7: Changed accordingly (Page 11, Line 453).

Point 8: Please, review citation number 1.

Response 8: The citation has been revised to "Godfray, H. C.; Beddington, J. R.; Crute, I. R.; Haddad, L.; Lawrence, D.; Muir, J. F.; Pretty, J.; Robinson, S.; Thomas, S. M.; Toulmin, C., Food security: the challenge of feeding 9 billion people. Science 2010, 327, (5967), 812-8" (Page 12, Lines 500-501).

Point 9: Could you add a sentence or a paragraph about future research lines in the Conclusions Section?

Response 9: As you mentioned, we believe that future research will mainly focus on three directions, so, I added this sentence at the Conclusion section These research findings promote reducing chemical pesticides, improving the effect and developing new strategies for insecticides resistance management in BPH. But there is still a lot of work to be done in the future. Firstly, the molecular mechanism of PZ regulation of rice resistance against brown planthopper is still not clear, analyzing this mechanism contributes to the development of synergists and the breeding of insect resistant varieties. Besides, the physiological changes mechanism of brown planthopper caused by PZ is also one of the future research directions. Thirdly, relevant field experiments also need to be conducted and the screening of other plant growth regulators that can serve as insecticide synergists need to be further studied (Page 11, Lines 466-475).

Round 2

Reviewer 1 Report

The authors addressed all the questions and comments made by the reviewer. The manuscript seems now in conditions to be accepted for publication in Antioxidants journal.

The English writting in fine in general.